# Curriculum learning for multilevel budgeted combinatorial problems

**Adel Nabli**     **Margarida Carvalho**
CIRRELT and Département d'Informatique et de Recherche Opérationnelle
Université de Montréal
`adel.nabli@umontreal.ca`
`carvalho@iro.umontreal.ca`

## Abstract

Learning heuristics for combinatorial optimization problems through graph neural networks have recently shown promising results on some classic NP-hard problems. These are single-level optimization problems with only one player. Multilevel combinatorial optimization problems are their generalization, encompassing situations with multiple players taking decisions sequentially. By framing them in a multi-agent reinforcement learning setting, we devise a value-based method to learn to solve multilevel budgeted combinatorial problems involving two players in a zero-sum game over a graph. Our framework is based on a simple curriculum: if an agent knows how to estimate the value of instances with budgets up to $B$, then solving instances with budget $B + 1$ can be done in polynomial time regardless of the direction of the optimization by checking the value of every possible afterstate. Thus, in a bottom-up approach, we generate datasets of heuristically solved instances with increasingly larger budgets to train our agent. We report results close to optimality on graphs up to $100$ nodes and a $185\times$ speedup on average compared to the quickest exact solver known for the Multilevel Critical Node problem, a max-min-max trilevel problem that has been shown to be at least $\Sigma_2^p$-hard.

## 1   Introduction

The design of heuristics to tackle real-world instances of NP-hard combinatorial optimization problems over graphs has attracted the attention of many Computer Scientists over the years [29]. With advances in Deep Learning [30] and Graph Neural Networks [63], the idea of leveraging the recurrent structures appearing in the combinatorial objects belonging to a distribution of instances of a given problem to learn efficient heuristics with a Reinforcement Learning (RL) framework has received an increased interest [6, 48]. Although these approaches show promising results on many fundamental NP-hard problems over graphs, such as Maximum Cut [4] or the Traveling Salesman Problem [37], the range of combinatorial challenges on which they are directly applicable is still limited.

Indeed, most of the combinatorial problems over graphs solved heuristically with Deep Learning [2, 4, 5, 13, 15, 37, 41, 47] are classic NP-hard problems for which the canonical optimization formulation is a single-level Mixed Integer Linear Program: there is one decision-maker[1] seeking to minimize a linear cost subject to linear constraints and integer requirements. However, in many real-world situations, decision-makers interact with each other. A particular case of such setting are sequential games with a hierarchy between players: an upper level authority (a leader) optimizes its goal subject to the response of a sequence of followers seeking to optimize their own objectives given

the actions previously made by others higher in the hierarchy. These problems are naturally modeled as Multilevel Programming problems (MPs) and can be seen as a succession of nested optimization tasks, *i.e.* mathematical programs with optimization problems in the constraints [9, 12, 65].

Thus, finding an optimal strategy for the leader in the multilevel setting may be harder than for single-level problems as evaluating the cost of a given strategy might not be possible in polynomial time: it requires solving the followers optimization problems. In fact, even Multilevel Linear Programming with a sequence of $L + 1$ players (levels) is $\Sigma_L^p$-hard [8, 17, 34]. In practice, exact methods capable to tackle medium-sized instances in reasonable time have been developed for max-min-max Trilevels, min-max Bilevels and more general Bilevel Programs (*e.g.*, [14, 22, 23, 46, 59]).

Despite the computational challenges intrinsic to MPs, these formulations are of practical interest as they properly model hierarchical decision problems. Originally appearing in economics in the bilevel form, designated as *Stackelberg competitions* [62], they have since been extended to more than two agents and seen their use explode in Operations Research [38, 57]. Thus, research efforts have been directed at finding good quality heuristics to solve those problems, *e.g.* [16, 24, 26, 60]. Hence, one can ask whether we can make an agent learn how to solve a wide range of instances of a given multilevel problem, extending the success of recent Deep Learning approaches on solving single-level combinatorial problems to higher levels.

In this paper, we propose a simple curriculum to learn to solve a common type of multilevel combinatorial optimization problem: budgeted ones that are zero-sum games played in a graph. Although the framework we devise is set to be general, we center our attention on the Multilevel Critical Node problem (MCN) [1] and its variants. The reasons for such a choice are manifold. First, the MCN is an example of a Defender-Attacker-Defender game [10] which received much attention lately as it aims to find the best preventive strategies to defend critical network infrastructures against malicious attacks. As it falls under the global framework of network interdiction games, it is also related to many other interdiction problems with applications ranging from floods control [51] to the decomposition of matrices into blocks [27]. Moreover, an exact method to solve the problem has been presented in [1] along with a publicly available dataset of solved instances[2], which we can use to assess the quality of our heuristic. Lastly, complexity results are available for several variants and sub-problems of MCN, indicating its challenging nature [49].

**Contributions.** Our contribution rests on several steps. First, we frame generic *Multilevel Budgeted Combinatorial problems* (MBC) as *Alternating Markov Games* [43, 44]. This allows us to devise a first algorithm, MultiL-DQN, to learn $Q$-values. By leveraging both the combinatorial setting *(the environment is deterministic)* and the budgeted case *(the length of an episode is known in advance)*, we motivate a curriculum, MultiL-Cur. Introducing a Graph Neural Networks based agent, we empirically demonstrate the efficiency of our curriculum on 3 versions of the MCN, reporting results close to optimality on graphs of size up to 100.

**Paper structure.** Section 2 formalizes the MBC problem. In Section 3, we provide an overview of the relevant literature. The MBC is formulated within the Multi-Agent RL setting in Section 4 along with the presentation of our algorithmic approaches: MultiL-DQN and MultiL-Cur. Section 5 states the particular game MCN in which our methodology is validated in Section 6.

## 2   Problem statement

The general setting for the MPs we are considering is the following: given a graph $G = (V, A)$, two concurrent players, the *leader* and the *follower*, compete over the same combinatorial quantity $S$, with the *leader* aiming to maximize it and the *follower* to minimize it. They are given a total number of moves $L \in \mathbb{N}$ and a sequence of budgets $(b_1, ..., b_L) \in \mathbb{N}^L$. Although our study and algorithms also apply to general integer cost functions $c$, for the sake of clarity, we will only consider situations where the cost of a move is its cardinality. We focus on perfect-information games, *i.e.* both players have full knowledge of the budgets allocated and previous moves. The *leader* always begins and the last move is attributed by the parity of $L$. At each turn $l \in [\![1, L]\!]$, the player concerned makes a set of $b_l$ decisions about the graph. This set is denoted by $A_l$ and constrained by the previous moves $(A_1, .., A_{l-1})$. We consider games where players can only improve their objective by taking a decision: there is no incentive to pass. Without loss of generality, we can assume that $L$ is odd. Then,

the Multilevel Budgeted Combinatorial problem (MBC) can be formalized as:

$$\text{(MBC)} \qquad \max_{|A_1| \le b_1} \min_{|A_2| \le b_2} ... \max_{|A_L| \le b_L} S(G, A_1, A_2, ..., A_L). \qquad (1)$$

MBC is a zero-sum game as both leader and follower have the same objective function but their direction of optimization is opposite. A particular combinatorial optimization problem is defined by specifying the quantity $S$, fixing $L$, and by characterizing the nature of both the graph *(e.g directed, weighted)* and of the actions allowed at each turn *(e.g labeling edges, removing nodes)*. The problem being fixed, a distribution $\mathbb{D}$ of instances $i \sim \mathbb{D}$ is determined by setting a sampling law for random graphs and for the other parameters, having specified bounds beforehand: $n = |V| \in [\![n^{min}, n^{max}]\!]$, $|A| \in [\![d^{min} \times n(n-1), d^{max} \times n(n-1)]\!]$, $(b_1, ..., b_L) \in [\![b_1^{min}, b_1^{max}]\!] \times ... \times [\![b_L^{min}, b_L^{max}]\!]$. Our ultimate goal is thus to learn good quality heuristics that manage to solve each $i \sim \mathbb{D}$.

In order to achieve that, we aim to leverage the recurrent structures appearing in the combinatorial objects in the distribution $\mathbb{D}$ by learning graph embeddings that could guide the decision process. As data is usually very scarce (datasets of exactly solved instances being hard to produce), the go-to framework to learn useful representations in these situations is Reinforcement Learning [58].

## 3 Related Work

The combination of graph embedding with reinforcement learning to learn to solve distributions of instances of combinatorial problems was introduced by Dai *et al.* [15]. Thanks to their S2V-DQN meta-algorithm, they managed to show promising results on three classic budget-free NP-hard problems. Thenceforth, there is a growing number of methods proposed to either improve upon S2V-DQN results [4, 13, 37, 41, 47] or tackle other types of NP-hard problems on graphs [2]. As all these approaches focus on single player games, they are not directly applicable to MBC.

To tackle the multiplayer case, Multi-Agent Reinforcement Learning (MARL) [42, 55] appears as the natural toolbox. The combination of Deep Learning with RL recently led to one of the most significant breakthrough in perfect-information, sequential two-player games: AlphaGo [56]. Although neural network based agents managed to exceed human abilities on other combinatorial games (*e.g.* backgammon [61]), these approaches focus on one fixed board game. Thus, they effectively learn to solve only one (particularly hard) instance of a combinatorial problem, whereas we aim to solve a whole distribution of them. Hence, the MBC problem we propose to study is at a crossroads between previous works on MARL and deep learning for combinatorial optimization.

An MBC with $L$ levels is potentially $\Sigma_L^p$-hard, and hence, it can be challenging to solve. One of the strategies used to learn to solve complex problems is to break them down in a sequence of learning tasks that are increasingly harder, a concept known as Curriculum Learning [7]. For supervised learning, Bengio *et al.* [7] showed that gradually increasing the entropy of the training distribution helped. However in RL, breaking down a task in sub-problems that can be ordered by difficulty is non trivial [31]. In robotics, [25, 33] proposed to start from the *goal* (*e.g.*, open a door) and give a starting state that is gradually further from that goal. These methods assume at least one known goal state that is used as a seed for expansion. For video games, [52] adapted the concept with a starting state increasingly further from the end of a demonstration. However, here, the goal is not to "reach a particular end state": there is no goal state at all. Rather, what we want is to "take an optimal decision at each level", and, particularly, take optimal decisions at the beginning of the game, *i.e.*, at the highest level of the multilevel optimization problem, given that all subsequent actions *will* be optimal. To do that, we show that we can start training our agent on the penultimate states of the sequential decision processes, where the optimization problems are easy to solve as there is only one unit of budget left, and, gradually, consider instances with larger budgets, which effectively corresponds to problems arriving earlier in the sequence of decisions, until we return to the original problem. Thus, contrary to [25, 33, 52], we do not "reverse time" to artificially build a sequence of tasks starting further from a goal state and, subsequently, harder to solve it in the hope of learning how to reach this goal from all possible starting states. Rather, we stack new optimization problems on top of previous ones, which gradually increases the computational complexity of the task, in order to learn to act optimally in optimization problems with an increasing number of levels.

Finally, taking another direction, some shifted their attention from specific problems to rather focus on general purpose solvers. For example, methodologies have been proposed to speed up the branch-and-bound implementations for (single-level) linear combinatorial problems by learning to branch [3]

using Graph Convolutional Neural Networks [28], Deep Neural Networks [64] or RL [19] to name some recent works; see the surveys [6, 45]. To the best of our knowledge, the literature on machine learning approaches for general multilevel optimization is restricted to the linear non-combinatorial case. For instance, in [32, 54] the linear multilevel problems are converted into a system of differential equations and solved using recurrent neural networks.

# 4   Multi-Agent Reinforcement Learning framework

Whereas single agent RL is usually described with Markov Decision Processes, the framework needs to be extended to account for multiple agents. This has been done in the seminal work of Shapley [53] by introducing Markov Games. In our case, we want to model two-player games in which moves are not played simultaneously but alternately. The natural setting for such situation was introduced by Littman in [42, 43, 44] under the name of *Alternating Markov Games*.

## 4.1   Alternating Markov Games

An Alternating Markov Game involves two players: a maximizer and a minimizer. It is defined by the tuple $\langle \mathcal{S}_1, \mathcal{S}_2, \mathcal{A}_1, \mathcal{A}_2, P, R \rangle$ with $\mathcal{S}_i$ and $\mathcal{A}_i$ the set of states and actions, respectively, for player $i$, $P$ the transition function mapping state-actions pairs to probabilities of next states and $R$ a reward function. For $s \in \mathcal{S} = \mathcal{S}_1 \cup \mathcal{S}_2$, we define $V^*(s)$ as the expected reward of the concerned agent for following the optimal minimax policy against an optimal opponent starting from state $s$. In a similar fashion, $Q^*(s, a)$ is the expected reward for the player taking action $a$ in state $s$ and both agents behaving optimally thereafter. Finally, with the introduction of the discount factor $\gamma$, we can write the generalized Bellman equations for Alternating Markov Games [43]:

$$V^*(s) = \begin{cases} \max_{a_1 \in \mathcal{A}_1} Q^*(s, a_1) & \text{if } s \in \mathcal{S}_1 \\ \min_{a_2 \in \mathcal{A}_2} Q^*(s, a_2) & \text{otherwise} \end{cases} \tag{2}$$

$$Q^*(s, a) = R(s, a) + \gamma \sum_{s'} P(s, a, s') V^*(s'). \tag{3}$$

## 4.2   MARL formulation of the Multilevel Budgeted Combinatorial problem

We now have all the elements to frame the MBC in the Alternating Markov Game framework. The *leader* is the maximizer and the *follower* the minimizer. The states $s_t$ consist of a graph $G_t$ and a tuple of budgets $\mathcal{B}_t = (b_1^t, ..., b_L^t) = (0, ..., 0, k, b_{l+1}, ..., b_L)$ with $k \leq b_l$, beginning with $s_0 \sim \mathbb{D}$. Thus, the value function is defined with:

$$V^*(s_0) = \max_{|A_1| \leq b_1^0} \min_{|A_2| \leq b_2^0} ... \max_{|A_L| \leq b_L^0} S(G_0, A_1, A_2, ..., A_L). \tag{4}$$

The game is naturally sequential with an episode length of $L$: each time step $t$ corresponds to a level $l \in [\![1, L]\!]$. The challenge of such formulation is the size of the action space that can become large quickly. Indeed, in a graph $G$ with $n$ nodes, and $leader$'s budget $b_1$, if the action that he/she can perform is *"removing a set of nodes from the graph"* (a common move in network interdiction games), then the size of the action space for the first move of the game is $\binom{n}{b_1}$. To remedy this, we redefine what is a step in the sequential decision process. We define $\mathcal{A}_1, ..., \mathcal{A}_L$ the sets of *individual* decisions available at each level $l$. Then, we make the simplifying observation that a player making a *set* of $b_l$ decisions $A_l$ in one move is the same as him/her making a *sequence* of $b_l$ decisions $(a_l^1, ..., a_l^{b_l}) \in \mathcal{A}_l \times (\mathcal{A}_l \backslash \{a_l^1\}) \times ... \times (\mathcal{A}_l \backslash \{a_l^1, .., a_l^{b_l-1}\})$ in one strike. More formally, we have the simple lemma *(proof in Appendix A.1)*:

**Lemma 4.1.** *The Multilevel Budgeted Combinatorial optimization problem (1) is equivalent to:*

$$\max_{a_1^1 \in \mathcal{A}_1} ... \max_{a_1^{b_1} \in \mathcal{A}_1 \backslash \{a_1^1, .., a_1^{b_1-1}\}} \min_{a_2^1 \in \mathcal{A}_2} ... \max_{a_L^{b_L} \in \mathcal{A}_L \backslash \{a_L^1, .., a_L^{b_L-1}\}} S(G, \{a_1^1, .., a_1^{b_1}\}, .., \{a_L^1, .., a_L^{b_L}\}).$$

In this setting, the length of an episode is no longer $L$ but $B = b_1 + ... + b_L$: the *leader* makes $b_1$ sequential actions, then the *follower* the $b_2$ following ones, etc. To simplify the notations, we

re-define the $\mathcal{A}_t$ as the sets of actions available for the agent playing at time $t$. As each action takes place on the graph, $\mathcal{A}_t$ is readable from $s_t$. Moreover, we now have $|\mathcal{A}_t| = \mathcal{O}(|V| + |A|)$.

The environments considered in MBC are deterministic and their dynamics are completely known. Indeed, given a graph $G_t$, a tuple of budgets $\mathcal{B}_t = (b_1^t, .., b_L^t)$ and a chosen action $a_t \in \mathcal{A}_t$ *(e.g removing the node $a_t$)*, the subsequent graph $G_{t+1}$, tuple of budgets $\mathcal{B}_{t+1}$ and next player are completely and uniquely determined. Thus, we can introduce *the next state function $N$* that maps state-action couples $(s_t, a_t)$ to the resulting afterstate $s_{t+1}$, and $p$ as the function that maps the current state $s_t$ to the player $p \in \{1, 2\}$ whose turn it is to play. As early rewards weight the same as late ones, we set $\gamma = 1$. Finally, we can re-write equations (2) and (3) as:

$$V^*(s_t) = \begin{cases} \max_{a_t \in \mathcal{A}_t} \left( R(s_t, a_t) + V^*(N(s_t, a_t)) \right) & \text{if } p(s_t) = 1 \\ \min_{a_t \in \mathcal{A}_t} \left( R(s_t, a_t) + V^*(N(s_t, a_t)) \right) & \text{otherwise} \end{cases} \qquad (5)$$

$$Q^*(s_t, a_t) = R(s_t, a_t) + V^*(N(s_t, a_t)). \qquad (6)$$

The definition of $R$ depends on the combinatorial quantity $S$ and the nature of the actions allowed.

## 4.3 Q-learning for the greedy policy

Having framed the MBC in the Markov Game framework, the next step is to look at established algorithms to learn $Q^*$ in this setup. Littman originaly presented *minimax Q-learning* [42, 43] to do so, but in matrix games, where all possible outcomes are enumerated. An extension using a neural network $\hat{Q}$ to estimate $Q$ has been discussed in [20]. However, their algorithm, Minimax-DQN, is suited for the simultaneous setting and not the alternating one. The main difference being that the former requires the extra work of solving a Nash game between the two players at each step, which is unnecessary in the later as a greedy policy exists [42]. To bring Minimax-DQN to the alternating case, we present MultiL-DQN, an algorithm inspired by S2V-DQN [15] but extended to the multilevel setting: we alternate between a $\min$ and a $\max$ in both the greedy rollout and the target definition. See Appendix B.1 for the pseudo-code.

## 4.4 A curriculum taking advantage of the budgeted combinatorial setting

With MultiL-DQN, the learning agent directly tries to solve instances drawn from $\mathbb{D}$, which can be very hard theoretically speaking. However, Lemma 4.1 shows that, at the finest level of granularity, MBC is actually made of $B$ nested sub-problems. As we know $B_{max} = b_1^{max} + ... + b_L^{max}$, the maximum number of levels considered in $\mathbb{D}$, instead of directly trying to learn the values of the instances from this distribution, we can ask whether beginning by focusing on the simpler sub-problems and gradually build our way up to the hard ones would result in better final results.

This reasoning is motivated by the work done by Bengio *et al.* [7] on Curriculum Learning. Indeed, it has been shown empirically that breaking down the target training distribution into a sequence of increasingly harder ones actually results in better generalization abilities for the learning agent. But, contrary to the *continuous* setting devised in their work, where the parameter governing the hardness *(entropy)* of the distributions considered is continuously varying between $0$ and $1$, here we have a natural *discrete* sequence of increasingly harder distributions to sample instances from.

Indeed, our ultimate goal is to learn an approximate function $\hat{Q}$ to $Q^*$ *(or equivalently $\hat{V}$ to $V^*$ (6))* so that we can apply the subsequent greedy policy to take a decision. Thus, $\hat{Q}$ has to estimate the state-action values of every instance appearing in a sequence of $B$ decisions. Although the *leader* makes the first move on instances from $\mathbb{D}$, as our game is played on the graph itself, the distribution of instances on which the second decision is made is no longer $\mathbb{D}$ but *instances from $\mathbb{D}$ on which a first optimal action for the leader has been made*. If we introduce the function

$$\begin{aligned} N_{l,\pi^*}^k : \quad & \mathcal{S} \longrightarrow \mathcal{S} \\ & s_t = (G_t, \mathcal{B}_t) \mapsto \begin{cases} N(s_t, a_t^* \sim \pi^*(\mathcal{A}_t)) & \text{if } \mathcal{B}_t = (0, .., 0, k, b_{l+1}, .., b_L) \\ s_t & \text{otherwise} \end{cases} \end{aligned} \qquad (7)$$

then, from top to bottom, we want $\hat{Q}$ to estimate the values of taking an action starting from the states $\mathbb{D}_1^* = \mathbb{D}$ where the first action is made, then $\mathbb{D}_2^* = N_{1,\pi^*}^{b_1^{max}}(\mathbb{D})$ where the second action is made on instances with original budget $b_1^{max}$ at level 1, and all the way down to

$$\mathbb{D}_{B_{max}}^* = N_{L,\pi^*}^2 \circ ... \circ N_{L,\pi^*}^{b_L^{max}} \circ ... \circ N_{1,\pi^*}^{b_1^{max}-1} \circ N_{1,\pi^*}^{b_1^{max}}(\mathbb{D}). \qquad (8)$$

As the maximum total budget is $B_{max}$, $\hat{Q}$ has to effectively learn to estimate values from $B_{max}$ different distributions of instances, one for each possible budget in $[\![1, b_l^{max}]\!]$ for each $l \in [\![1, L]\!]$. But the instances in these distributions are not all equally hard to solve. Actually, the tendency is that the deeper in the sequence of decisions a distribution is, the easier to solve are the instances sampled

---

**Algorithm 1:** MultiL-Cur

1  Initialize the value-network $\hat{V}$ with weights $\hat{\theta}$ ;
2  Initialize the list of experts $\mathbb{L}_{\hat{V}}$ to be empty ;
3  **for** $j = B_{max}, ..., 2$ **do**
4      Create $\mathcal{D}_{train}^j, \mathcal{D}_{val}^j$ by sampling $(s_j^r \sim \mathbb{D}_j^r, \texttt{GreedyRollout}^a(s_j^r, \mathbb{L}_{\hat{V}}))$;
5      Initialize $\hat{V}_j$, the expert of level $j$ with $\hat{\theta}_j = \hat{\theta}$ ;
6      Initialize the loss on the validation set $\mathcal{L}_{val}^j$ to $\infty$ ;
7      **for** *epoch* $e = 1, ..., E$ **do**
8          **for** *batches* $(s_i, \hat{y}_i)_{i=1}^m \in \mathcal{D}_{train}^j$ **do**
9              Update $\hat{\theta}$ over $\frac{1}{m} \sum_{i=1}^m \left( \hat{y}_i - \hat{V}(s_i) \right)^2$ with Adam [35] ;
10             **if** *number of new updates* $= T_{val}$ **then**
11                 **if** $\mathcal{L}_{val}^{new} = \frac{1}{N_{val}} \sum_{k=1}^{N_{val}} \left( \hat{y}_k - \hat{V}(s_k) \right)^2 < \mathcal{L}_{val}^j$ **then**
12                     $\hat{\theta}_j \leftarrow \hat{\theta}$ ; $\mathcal{L}_{val}^j \leftarrow \mathcal{L}_{val}^{new}$ ;
13     Add $\hat{V}_j$ to $\mathbb{L}_{\hat{V}}$
14 **return** *the trained list of experts* $\mathbb{L}_{\hat{V}}$

---

[a] The pseudo-code for the `GreedyRollout` function is available in Appendix B.2

from it. For example, the last distribution $\mathbb{D}_{B_{max}}^*$ contains all the instances with a total remaining budget of at most 1 that it is possible to obtain for the last move of the game when every previous action was optimal. The values of these instances can be computed exactly in polynomial time[3] by checking the reward obtained with every possible action. Thus, if we had access to the $\{D_j^*\}_{j=1}^{B_{max}}$, then a natural curriculum would be to begin by sampling a dataset of instances $s_{B_{max}}^* \sim \mathbb{D}_{B_{max}}^*$, find their exact value in polynomial time, and train a neural network $\hat{V}$ on the couples $(s_{B_{max}}^*, V^*(s_{B_{max}}^*))$. Once this is done, we could pass to the $s_{B_{max}-1}^* \sim \mathbb{D}_{B_{max}-1}^* = N_{L,\pi^*}^3 \circ ... \circ N_{1,\pi^*}^{b_1^{max}} (\mathbb{D})$. As these instances have a total budget of at most 2, we can heuristically solve them by generating every possible afterstate and, by using the freshly trained $\hat{V}$, take a greedy decision to obtain their approximate targets. In a bottom up approach, we could continue until $\hat{V}$ is trained on the $B_{max}$ different distributions. The challenge of this setting being that we do not know $\pi^*$ and hence, $\{\mathbb{D}_j^*\}_{j=1}^{B_{max}}$ are not available. To remedy this, we use a proxy, $\mathbb{D}_j^r$, obtained by following the random policy $a_t^r \sim \mathcal{U}(\mathcal{A}_t)$ for the sequence of previous moves, *i.e.*, we use $N_{l,\pi^r}^k$ instead of $N_{l,\pi^*}^k$. Doing so still allows us to learn the values of the instances we actually care about *(proof in Appendix A.2)*:

**Lemma 4.2.** $\forall j \in [\![2, B_{max}]\!]$, $\mathrm{supp}(\mathbb{D}_j^*) \subseteq \mathrm{supp}(\mathbb{D}_j^r)$.

Thus, by learning the value of instances sampled from $\mathbb{D}_j^r$, we also learn values of instances from $\mathbb{D}_j^*$. To avoid the pitfall of catastrophic forgetting [39] happening when a neural network switches of training distribution, each time it finishes to learn from a $\mathbb{D}_j^r$ and before the transition $j$ to $j-1$, we freeze a copy of $\hat{V}$ and save it in memory as an *"expert of level $j$"*. Hence, at level $l \in [\![1, L]\!]$ with budget $k_l \in [\![1, b_l^{max}]\!]$, we have access to a list of trained experts $\mathbb{L}_{\hat{V}}$ that can take decisions for each next step in the sequences, corresponding to instances with either a lower level $l' < l$ or a lower budget $k_{l'} < k_l$. To train $\hat{V}$, we first sample a dataset $\{s_0^{(i)}\}_{i=1}^N$ of instances from $\mathbb{D}$. Then, for each instance, we take a sequence of random decisions to arrive at level $l$ and budget at most $k_l$ by applying the operator $N_{l,\pi^r}^{k_l+1} \circ ... \circ N_{1,\pi^r}^{b_1^{max}-1} \circ N_{1,\pi^r}^{b_1^{max}}$ to the $s_0^{(i)}$. Finally, to retrieve the approximate targets of the subsequent instances that we will use to train the current $\hat{V}$, we generate every possible afterstate and use the previously trained $\mathbb{L}_{\hat{V}}$. The complete procedure is formalized in Algorithm 1.

## 5 The Multilevel Critical Node Problem

The MCN [1] is a trilevel budgeted combinatorial problem on a weighted graph $G = (V, A)$. The *leader* is called the *defender* and the *follower* is the *attacker*. The *defender* begins by removing (vaccinate) a set $D$ of $\Omega$ nodes, then the *attacker* labels a set $I$ of $\Phi$ nodes as attacked (infected), and finally the *defender* removes (protects) a set $P$ of $\Lambda$ new nodes that were not attacked. Once all the

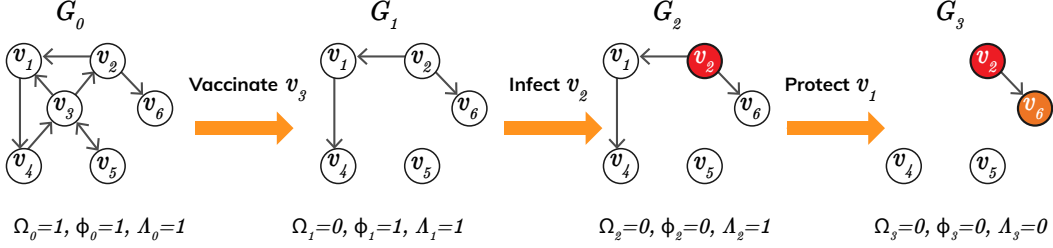

Figure 1: Example of an MCN game on a directed graph with unitary weights and initial budgets $\Omega = \Phi = \Lambda = 1$. Here, $S = 4$, $\{v_1, v_3, v_4, v_5\}$ are saved while $\{v_2, v_6\}$ are infected in the end.

moves are done, attacked nodes are the source of a cascade of infections that propagate through arcs from node to node. All nodes that are not infected are saved. As $D$ and $P$ were removed from the graph, they are automatically saved and the *leader* receives $w(D)$ and $w(P)$, the weights of the nodes in $D$ and $P$, as reward when performing those actions. The quantity $S$ that the *defender* seeks to maximize is thus the *sum of the weights of the saved nodes in the end of the game*, while the *attacker* aims to minimize it. An example of a game is presented in Figure 1. The problem can be written as:

$$\max_{\substack{D \subseteq V \\ |D| \leq \Omega}} \quad \min_{\substack{I \subseteq V \setminus D \\ |I| \leq \Phi}} \quad \max_{\substack{P \subseteq V \setminus (D \cup I) \\ |P| \leq \Lambda}} \quad S(G, D, I, P). \tag{9}$$

With unit weights, the MCN has been shown to be at least NP-hard on undirected graphs and at least $\Sigma_2^p$-hard on directed ones. In the more general version, with positive weights and costs associated to each node, the problem is $\Sigma_3^p$-complete [49].

## 6 Computational Results

**Instances.** We studied 3 versions of the MCN: undirected with unit weights (MCN), undirected with positive weights (MCN$_w$), and directed with unit weights (MCN$_{dir}$). The first distribution of instances considered is $\mathbb{D}^{(1)}$, constituted of Erdos-Renyi graphs [18] with size $|V|^{(1)} \in [\![10, 23]\!]$ and arc density $d^{(1)} \in [0.1, 0.2]$. For the weighted case, we considered integer weights $w \in [\![1, 5]\!]$. The second distribution of instances $\mathbb{D}^{(2)}$ focused on larger graphs with $|V|^{(2)} \in [\![20, 60]\!]$, $d^{(2)} \in [0.05, 0.15]$. To compare our results with exact ones, we used the budgets reported in the experiments of the original MCN paper [1]: $\Omega \in [\![0, 3]\!]$, $\Phi \in [\![1, 3]\!]$ and $\Lambda \in [\![0, 3]\!]$.

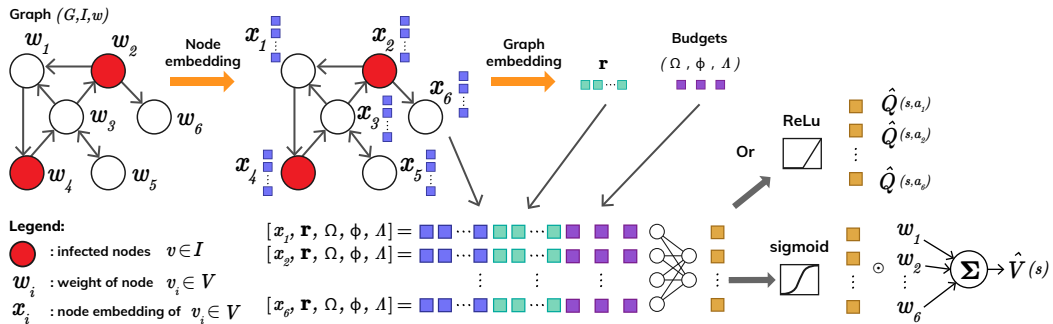

Figure 2: Architecture of the two neural networks used: $\hat{V}$ and $\hat{Q}$. $\hat{V}$ computes a score $\in [0, 1]$ for each node, which can be interpreted as its probability to be saved given the context (graph embedding and budgets).

**Graph embedding.** The architectures presented in Figure 2 was implemented with Pytorch Geometric [21] and Pytorch 1.4 [50]. At the beginning, a node $v \in V$ has only 2 features $(w_v, \mathbb{1}_{v \in I})$: its weight and an indicator of whether it is attacked or not. The first step of our node embedding method is to concatenate the node's two features with its *Local Degree Profile* [11] consisting in 5 simple statistics on its degree. Following the success of Attention on routing problems reported in [37], we then apply their Attention Layer. As infected nodes are the ones in the same connected component as

attacked ones in the graph, we sought to propagate the information of each node to all the others it is connected to. That way, the *attacker* could know which nodes are already infected before spending the rest of his/her budget, and the *defender* could realize which nodes are to protect in his/her last move. So, after the Attention Layers, we used an APPNP layer [36] that, given the matrix of nodes embedding $\mathbf{X}^{(0)}$, the adjacency matrix with inserted self-loops $\hat{\mathbf{A}}$, $\hat{\mathbf{D}}$ its corresponding diagonal degree matrix, and a coefficient $\alpha \in [0, 1]$, recursively applies $K$ times:

$$\mathbf{X}^{(k)} = (1 - \alpha)\hat{\mathbf{D}}^{-1/2}\hat{\mathbf{A}}\hat{\mathbf{D}}^{-1/2}\mathbf{X}^{(k-1)} + \alpha\mathbf{X}^{(0)}. \tag{10}$$

To achieve our goal, the value of $K$ must be at least equal to the size of the largest connected component possible to have in the distribution of instances $\mathbb{D}^{(i)}$. We thus used $K^{(i)} = \max(|V|^{(i)})$. Finally, the graph embedding method we used is the one presented in [40]. Given two neural networks $h_{gate}$ and $h_r$ which compute, respectively, a score $\in \mathbb{R}$ and a projection to $\mathbb{R}^r$, the graph level representation vector it outputs is:

$$\mathbf{r} = \sum_{i=1}^{n} \text{softmax}(h_{gate}(\mathbf{x_i})) \odot h_r(\mathbf{x_i}). \tag{11}$$

To train our agent and at inference, we used one gpu of a cluster of NVidia V100SXM2 with 16G of memory[4]. Further details of the implementation are discussed in Appendix D.

**Algorithms.** We compared our algorithms on $\mathbb{D}^{(1)}$ and trained our best performing one on $\mathbb{D}^{(2)}$. But as it is, comparing MultiL-DQN with MultiL-Cur may be unfair. Indeed, MultiL-DQN uses a $Q$-network whereas MultiL-Cur uses a value network. The reason why we used $\hat{V}$ instead of $\hat{Q}$ in our second algorithm are twofold. First, as our curriculum leans on the abilities of experts trained on smaller budgets to create the next training dataset, computing *values* of afterstates is necessary to heuristicaly solve instances with larger budgets. Second, as MCN is a game with one player removing nodes from the graph, symmetries can be leveraged in the afterstates. Indeed, given the graph $G'$ resulting of a node deletion, many couples $(G, v)$ of graph and node to delete could have resulted in $G'$. Thus, $\hat{Q}$ has to learn that all these possibilities are similar, while $\hat{V}$ only needs to learn the value of the shared afterstate, which is more efficient [58]. To fairly compare the algorithms, we thus introduce MultiL-MC, a version of MultiL-DQN based on a value network and using Monte-Carlo samples as in MultiL-Cur. Its pseudo-code is available in Appendix B.3.

Table 1: Evolution during training of the loss on 8 test sets of 1000 exactly solved instances $\in \mathbb{D}^{(1)}$. Averaged on 3 runs. We measured the loss on distributions arriving at different stages of the curriculum. The approximation ratio and optimality gap were measured after training and averaged over all the tests sets.

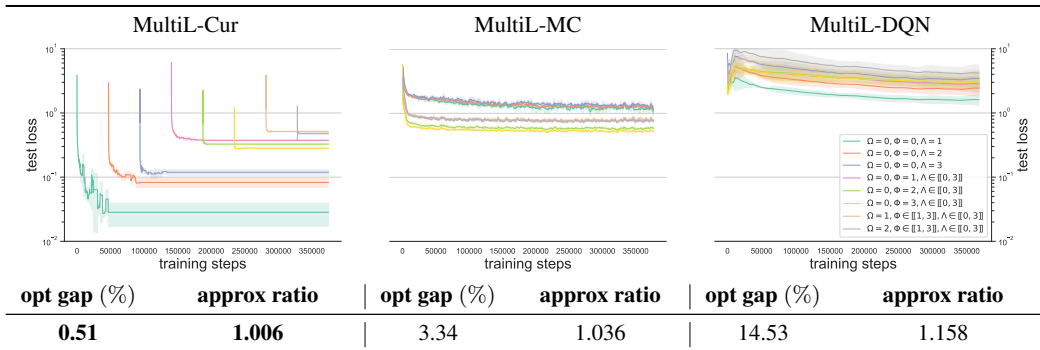

| opt gap (%) | approx ratio | opt gap (%) | approx ratio | opt gap (%) | approx ratio |
|---|---|---|---|---|---|
| **0.51** | **1.006** | 3.34 | 1.036 | 14.53 | 1.158 |

**Baselines.** Results from Table 1 indicate that MultiL-Cur is the best performing algorithm on $\mathbb{D}^{(1)}$. Thus, we trained our learning agent with it on $\mathbb{D}^{(2)}$ and tested its performance on the datasets generated in [1]. We compare the results with 2 other heuristics: the random policy *(for each instance, we average the value given by* 10 *random episodes)*, and the DA-AD heuristic [1]. The latter consists in separately solving the two bilevel problems inside MCN: $D$ is chosen by setting $\Lambda$ to 0 and exactly

solving the *Defender-Attacker* problem, while $I$ and $P$ are determined by solving the subsequent *Attacker-Defender* problem. The metrics we use are the optimality gap $\eta$ and the approximation ratio $\zeta$. Given $n_i$, the number of instances of a said type for which the optimal value $v^*$ is available, $\eta = \frac{1}{n_i} \sum_{k=1}^{n_i} \frac{|v_k^* - \hat{v}_k|}{v_k^*}$ and $\zeta = \frac{1}{n_i} \sum_{k=1}^{n_i} \max(\frac{v_k^*}{\hat{v}_k}, \frac{\hat{v}_k}{v_k^*})$. In Table 2, we report the inference times $t$ in seconds for our trained agents. The ones for the exact method and DA-AD are from [1].

Table 2: Comparison between several heuristics and exact methods. Results on MCN are computed on the dataset of the original paper [1]. For $\text{MCN}_{dir}$ and $\text{MCN}_w$, we generated our own datasets by making small adaptations to the exact solver of [1] originally suited for MCN, see Appendix C for details.

| | | | | **MCN** | | | | | | **$\text{MCN}_{dir}$** | | **$\text{MCN}_w$** | |
| | exact | random | | DA-AD | | | cur | | | cur | | cur | |
| $\|V\|$ | $t(s)$ | $\eta(\%)$ | $\zeta$ | $t(s)$ | $\eta(\%)$ | $\zeta$ | $t(s)$ | $\eta(\%)$ | $\zeta$ | $\eta(\%)$ | $\zeta$ | $\eta(\%)$ | $\zeta$ |
|---|---|---|---|---|---|---|---|---|---|---|---|---|---|
| 20 | 29 | 68 | 3.32 | 6 | **0.3** | **1.00** | **0.4** | 0.5 | 1.00 | 5.7 | 1.07 | 6.9 | 1.07 |
| 40 | 241 | 52 | 2.64 | 13 | 7.6 | 1.09 | **0.9** | 5.0 | **1.06** | 11.9 | 1.13 | 6.5 | 1.07 |
| 60 | 405 | 68 | 3.24 | 38 | 7.3 | 1.09 | **1.5** | 4.4 | **1.05** | 4.4 | 1.05 | 3.7 | 1.04 |
| 80 | 636 | 55 | 2.28 | 60 | 3.8 | 1.04 | **2.8** | 2.7 | **1.03** | 1.6 | 1.02 | 2.8 | 1.03 |
| 100 | 848 | 45 | 1.86 | 207 | **2.7** | **1.03** | **8.7** | 49.6 | 1.50 | 1.8 | 1.02 | 4.1 | 1.05 |

**Discussion.** Although the results in Table 1 are the outcome of a total of $\sim 900000$ episodes and $\sim 350000$ optimization steps for all 3 algorithms, our experiments show that we can divide by 2 the data and 4 the number of steps without sacrificing much the results on $\mathbb{D}^{(1)}$ for the curriculum, which cannot be said of MultiL-DQN that is data hungry, see Appendix E for details. The major drawback of MultiL-Cur is that it needs to compute all possible afterstates at each step of the rollout. This does not scale well with the graph's size: having 100 nodes for the first step means that there are 100 graphs of size 99 to check. Thus, the curriculum we present is a promising step towards automatic design of heuristis, while opening new research directions on restricting the exploration of rollouts.

Table 2 reveals that the results given by the MultiL-Cur algorithm are close to the optimum for a fraction of the time necessary to both DA-AD and the quickest exact solver known *(MCN$^{MIX}$, presented in [1])*. For the MCN instances, the jump in the metrics for graphs of size 100 is due to one outlier among the 85 exactly solved instances of this size. When removed, the values of $\eta$ and $\zeta$ drop to 17.8 and 1.18. The performances measured are consistent accross different problems as we also report low values of $\eta$ and $\zeta$ for $\text{MCN}_{dir}$ and $\text{MCN}_w$. The curriculum we devised is thus a robust and efficient way to train agents in a Multilevel Budgeted setting.

## Conclusion

In this paper, we proposed a new method for learning to solve Multilevel Budgeted Combinatorial problems by framing them in the Alternative Markov Game framework. To train our agent, we broke down the classical multilevel formulation into a sequence of unitary steps, allowing us to devise a curriculum leveraging the simple observation that the further down in the decision process a problem is, the lower its computational complexity is. Using a bottom up approach, we demonstrated that this learning strategy outperforms more classical RL algorithms on a trilevel problem, the MCN.
This study also restates the difficulty of scaling up methods for multilevel optimization due to their theoretical and empirical complexity. This motivates the development of heuristics that can both serve as stand-alone methods or warm-start exact solvers. However, the one we devised is based on an afterstate value function, raising scalability issues. But, our work being the first looking to such problems, it sets the ground for less expensive curricula, which we leave as a future direction.
Finally, we highlight the need for further study on the metrics necessary to properly report the optimality gaps in the multilevel setting. The main drawback of heuristics for multilevel optimization is on their evaluation: given a leader's strategy, its associated value can only be evaluated if the remaining levels are solved to optimality. This means that in opposition to single-level optimization, one must be very careful on the interpretation of the estimated reward associated with an heuristic method: we can be overestimating or underestimating it. In other words, it means that in the remaining levels, players might be failing to behave in an optimal way. Consequently, further research is necessary to provide guarantees on the quality of the obtained solution, namely, on dual bounds.

## Broader Impact

We propose a general framework for training agents to tackle a class of Multilevel Budgeted Combinatorial problems. Such models are widely used in Economics and Operations Research. In this study, we particularly focused on the Multilevel Critical Node problem (MCN). Regarding the usefulness of such problem for practical scenarios, the MCN could fit on several applications, *e.g.* to limit the fake news spread in social networks or in cyber security for the protection of a botnet against malware injections [1]. Thus, this could represent a step towards the design of more robust networks, but could also be used to identify their critical weaknesses for malicious agents. We do not anticipate that our work will advantage or disadvantage any particular group.

## Acknowledgments and Disclosure of Funding

The authors wish to thank the support of the Institut de valorisation des données and Fonds de Recherche du Québec through the FRQ–IVADO Research Chair in Data Science for Combinatorial Game Theory, and the Natural Sciences and Engineering Research Council of Canada through the discovery grant 2019-04557. This research was enabled in part by support provided by Calcul Québec (`www.calculquebec.ca`) and Compute Canada (`www.computecanada.ca`).

## Footnotes

[1]We will use interchangeably the words decision-maker, agent and player. Note that decision-maker, player and agent are usually used in Operations Research, Game Theory and Reinforcement Learning, respectively. Similarly for the words decision, strategy and policy.

[2] https://github.com/mxmmargarida/Critical-Node-Problem

[3] Assuming the quantity $S$ is computable in polynomial time.

[4]We make our code publicly available: `https://github.com/AdelNabli/MCN`

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
