[Supplementary Material]

# Appendices

## A  Proofs

**Lemma A.1.** *The Multilevel Budgeted Combinatorial optimization problem (1) is equivalent to:*

$$\max_{a_1^1 \in \mathcal{A}_1} ... \max_{a_1^{b_1} \in \mathcal{A}_1 \setminus \{a_1^1,..,a_1^{b_1-1}\}} \min_{a_2^1 \in \mathcal{A}_2} ... \max_{a_L^{b_L} \in \mathcal{A}_L \setminus \{a_L^1,..,a_L^{b_L-1}\}} S(G, \{a_1^1,..,a_1^{b_1}\},..,\{a_L^1,..,a_L^{b_L}\}).$$

*Proof.* We immediately have the following relation:

$$\max_{|A_1| \leq b_1} ... \max_{|A_L| \leq b_L} S(G, A_1, A_2,.., A_L) = \max_{a_1^1 \in \mathcal{A}_1} \max_{|A'_1| \leq b_1 - 1} ... \max_{|A_L| \leq b_L} S(G, \{a_1\} \cup A'_1, A_2,.., A_L)$$

As the same reasoning holds with $\min$, we can apply it recursively, which closes the proof. □

**Lemma A.2.** $\forall j \in [2, B_{max}], \operatorname{supp}(\mathbb{D}_j^*) \subseteq \operatorname{supp}(\mathbb{D}_j^r)$

*Proof.* For all $s_0 \sim \mathbb{D}$, for all $t \in [\![0, B-1]\!]$, we define $\mathcal{A}_t^*(a_0, ..., a_{t-1}) \subseteq \mathcal{A}_t(a_0, ..., a_{t-1})$ as the set of optimal actions at time $t$ in state $s_t$ for the player $p(s_t)$, where we made evident the dependence of $s_t$ on previous actions. As by assumption we consider games where players can only improve their objective by taking a decision, we have that $\forall t, \mathcal{A}_t \neq \emptyset \implies \mathcal{A}_t^* \neq \emptyset$. For a given $s_t$ and subsequent $\mathcal{A}_t$, recall that $a_t^r$ is defined as a random variable with values in $\mathcal{A}_t$ and following the uniform law. Given $s_0 \sim \mathbb{D}$, we take $(a_0^*, ..., a_{B-1}^*) \in \mathcal{A}_0^* \times ... \times \mathcal{A}_{B-1}^*(a_0^*, ..., a_{B-2}^*)$, one of the possible sequence of optimal decisions. Then, using the chain rule, it is easy to show by recurrence that $\forall t \in [\![0, B-1]\!], P(a_0^r = a_0^*, ..., a_t^r = a_t^*) > 0$. In words, every optimal sequence of decisions is generated with a strictly positive probability. □

## B  Algorithms

### B.1  MultiL-DQN

As the player currently playing is completely determined from $s_t$, we can use the same neural network $\hat{Q}$ to estimate all the state-action values, regardless of the player. We call $B_t$ the sum of all the budgets in $\mathcal{B}_t$ such that an episode stops when $B_t = 0$.

---
**Algorithm 2:** MultiL-DQN

1  Initialize the replay memory $\mathcal{M}$ to capacity $\mathcal{C}$ ;
2  Initialize the $Q$-network $\hat{Q}$ with weights $\hat{\theta}$ ;
3  Initialize the target-network $\tilde{Q}$ with weights $\tilde{\theta} = \hat{\theta}$ ;
4  **for** *episode* $e = 1, ..., E$ **do**
5      Sample $s_0 = (G_0, \mathcal{B}_0) \sim \mathbb{D}$ ;
6      $t \leftarrow 0$ ;
7      **while** $B_t \geq 1$ **do**
8          $a_t = \begin{cases} \text{random action } a_t \in \mathcal{A}_t & \text{w.p. } \epsilon \\ \arg\max_{a_t \in \mathcal{A}_t} \hat{Q}(s_t, a_t) & \text{otherwise if } p(s_t) = 1 \\ \arg\min_{a_t \in \mathcal{A}_t} \hat{Q}(s_t, a_t) & \text{otherwise if } p(s_t) = 2 \end{cases}$ ;
9          $s_{t+1} \leftarrow N(s_t, a_t)$ ;
10          $t \leftarrow t + 1$ ;
11          **if** $t \geq 1$ **then**
12              Add $(s_{t-1}, a_{t-1}, R(s_{t-1}, a_{t-1}), s_t)$ to $\mathcal{M}$ ;
13              Sample a random batch $\{(s_i, a_i, r_i, s_i')\}_{i=1}^m \overset{i.i.d}{\sim} \mathcal{M}$ ;
14              **for** $i = 1, .., m$ **do**
15                  $y_i = r_i + \mathbb{1}_{p(s_i')=1} \max_{a' \in \mathcal{A}'} \tilde{Q}(s_i', a') + \mathbb{1}_{p(s_i')=2} \min_{a' \in \mathcal{A}'} \tilde{Q}(s_i', a')$
16              Update $\hat{\theta}$ over $\frac{1}{m} \sum_{i=1}^m \left(y_i - \hat{Q}(s_i, a_i)\right)^2$ with Adam [8] ;
17              Update $\tilde{\theta} \leftarrow \hat{\theta}$ every $T_{target}$ steps
18  **return** *the trained Q-network* $\hat{Q}$

---

## B.2 Greedy Rollout

---
**Algorithm 3:** Greedy Rollout

---
**Input :** A state $s_t$ with total budget $B_t$ and a list of experts value networks $\mathbb{L}_{\hat{V}}$

**1** Initialize the value $\hat{v} \leftarrow 0$ ;

**2 while** $B_t \geq 1$ **do**

**3** $\quad$ Retrieve the expert of the next level $\hat{V}_{t+1}$ from the list $\mathbb{L}_{\hat{V}}$ ;

**4** $\quad$ Generate every possible afterstate $\mathcal{S}'_t \leftarrow \{N(s_t, a_t)\}_{a_t \in \mathcal{A}_t}$ ;

**5** $\quad s_{t+1} = \begin{cases} \arg\max_{s' \in \mathcal{S}'_t} \hat{V}_{t+1}(s') & \text{if } p(s_t) = 1 \\ \arg\min_{s' \in \mathcal{S}'_t} \hat{V}_{t+1}(s') & \text{if } p(s_t) = 2 \end{cases}$ ;

**6** $\quad \hat{v} \leftarrow \hat{v} + R(s_t, s_{t+1})$ ;

**7** $\quad t \leftarrow t + 1$ ;

**8 return** *the value* $\hat{v}$

---

## B.3 MultiL-MC

As we use Monte-Carlo samples as targets, the values of the targets sampled from the replay memory $\mathcal{M}$ is not dependent on the current expert as in DQN [12] but on a previous version of $\hat{V}$, which can become outdated quickly. Thus, to easily control the number of times an old estimate is used, we decided to perform an epoch on the memory every time $m$ new samples were pushed, and used a capacity $\mathcal{C} = k \times m$ so that the total number of times a Monte-Carlo sample is seen is directly $k$.

---
**Algorithm 4:** MultiL-MC

---
**1** Initialize the replay memory $\mathcal{M}$ to capacity $\mathcal{C}$ ;

**2** Initialize the value-network $\hat{V}$ with weights $\hat{\theta}$ ;

**3 for** *episode* $e = 1, ..., E$ **do**

**4** $\quad$ Sample $s_0 = (G_0, \mathcal{B}_0) \sim \mathbb{D}$ ;

**5** $\quad$ Initialize the memory of the episode $\mathcal{M}_e$ to be empty;

**6** $\quad$ Initialize the length of the episode $T \leftarrow 0$ ;

**7** $\quad$ **while** $B_t \geq 1$ **do** $\qquad\qquad\qquad\qquad$ // perform a Monte Carlo sample

**8** $\quad\quad a_t = \begin{cases} \text{random action } a_t \in \mathcal{A}_t & \text{w.p. } \epsilon \\ \arg\max_{a_t \in \mathcal{A}_t} \hat{V}(N(s_t, a_t)) & \text{otherwise if } p(s_t) = 1 \\ \arg\min_{a_t \in \mathcal{A}_t} \hat{V}(N(s_t, a_t)) & \text{otherwise if } p(s_t) = 2 \end{cases}$ ;

**9** $\quad\quad s_{t+1} = N(s_t, a_t)$ ;

**10** $\quad\quad$ Add $(s_t, R(s_t, a_t))$ to $\mathcal{M}_e$ ;

**11** $\quad\quad T \leftarrow T + 1$

**12** $\quad$ Initialize the target $y_T \leftarrow 0$ ;

**13** $\quad$ **for** $t = 1, ..., T$ **do** $\qquad\qquad\qquad\qquad$ // associate each state to its value

**14** $\quad\quad$ Recover $(s_{T-t}, R(s_{T-t}, a_{T-t}))$ from $\mathcal{M}_e$ ;

**15** $\quad\quad y_{T-t} \leftarrow y_{T-t+1} + R(s_{T-t}, a_{T-t})$ ;

**16** $\quad\quad$ Add $(s_{T-t}, y_{T-t})$ to $\mathcal{M}$

**17** $\quad$ **if** *there are more than* $m$ *new couples in* $\mathcal{M}$ **then**

**18** $\quad\quad$ Create a random permutation $\sigma \in \mathcal{S}_N$ ;

**19** $\quad\quad$ **for** *batches* $\{(s_i, y_i)\}_{i=1}^m \sim \sigma(\mathcal{M})$ **do** $\qquad$ // perform an epoch on the memory

**20** $\quad\quad\quad$ Update $\hat{\theta}$ over the loss $\frac{1}{m} \sum_{i=1}^m \left( y_i - \hat{V}(s_i) \right)^2$ with Adam [8]

**21 return** *the trained value-network* $\hat{V}$

---

## C Broadening the scope of the exact algorithm

In order to constitute a test set to compare the results given by our heuristics to exact ones, we used the exact method described in [2] to solve a small amount of instances. The algorithm they described was thought for the MCN problem, but is directly applicable without change on $\text{MCN}_{dir}$. However, in order to monitor the learning at each stage of the curriculum for MCN as in Table 1, there is a need to solve instances where node infections were already performed in the sequence of previous moves but there is still some budget left to spend for the attacker, which is not possible as it is in [2]. Moreover, small changes need to be made in order to solve instances of $\text{MCN}_w$.

## C.1 Adding nodes that are already infected

We denote by $J$ the set of nodes that are already infected at the attack stage and $\beta_v = \mathbb{1}_{v \in J}$ the indicator of whether node $v$ is in $J$ or not. Then, the total set of infected nodes after the attacker spend his/her remaining budget $\Lambda$ and infect new nodes $I$ is $J \cup I$. In order to find $I$, we use the AP algorithm of [2], with the following modification to the rlxAP optimization problem:

$$\min \qquad \Lambda p + \sum_{v \in V} \gamma_v$$

$$\sum_{v \in V} y_v \leq \Lambda$$

$$y_v \leq 1 - \beta_v \qquad \forall v \in V$$

$$h_v + \sum_{(u,v) \in A} q(u,v) - \sum_{(u,v) \in A} q(v,u) \geq 1 \qquad \forall v \in V$$

$$p - \sum_{(u,v) \in A} q(u,v) \geq 0 \qquad \forall v \in V$$

$$\gamma_v + |V| y_v - h_v \geq -|V| \beta_v \qquad \forall v \in V$$

$$p, \ h_v, \ \gamma_v, \ q(u,v) \geq 0 \qquad \forall v \in V, \ \forall (u,v) \in A$$

$$y_v \in \{0,1\} \qquad \forall v \in V$$

We indicated changes in blue. The notations for the variables being the ones from [2].

## C.2 Adding weights

Taking the weights $w_v$ of the nodes $v \in V$ into account in the optimization problems is even more straightforward. As the criterion to optimize is no longer the *number* of saved nodes but the *sum of their weights*, each time a cardinal of a set appears in the algorithms AP and MCN in [2], we replace it by the the sum of the weights of its elements. As for the optimization problems that are solved during the routines, we replace, in the Defender problem and in the 1lvlMIP:

$$\sum_{v \in V} \alpha_v \longrightarrow \sum_{v \in V} w_v \alpha_v$$

and in the rlxAP problem:

$$h_v + \sum_{(u,v) \in A} q(u,v) - \sum_{(u,v) \in A} q(v,u) \geq 1 \longrightarrow h_v + \sum_{(u,v) \in A} q(u,v) - \sum_{(u,v) \in A} q(v,u) \geq w_v$$

# D   Experiments details

## D.1   Architecture details

**Nodes embedding .**  The first step of the method described in Figure 2 is the node embedding part. Each node $v \in V$ begins with two features $\mathbf{x_v} = (w_v, \mathbb{1}_{v \in I})$: its weight and an indicator of whether it is attacked or not. First, we normalize the weights by dividing them with the sum of the weights in the graph such that each $w_v \in [0,1]$. We extend the two features with the *Local Degree Profile* of each node [3], which consists in 5 features on the degree:

$$\mathbf{x_v} = \mathbf{x_v} || (\deg(v), \min(DN(v)), \max(DN(v)), \mathrm{mean}(DN(v)), \mathrm{std}(DN(v))) \qquad (12)$$

with $\deg$ the degree of a node, and $DN$ the degrees of $\mathcal{N}(v)$ - the neighbors of $v$ in the graph. Then, we project our features $\mathbf{x_v} \in \mathbb{R}^7$ into $\mathbb{R}^{d_e}$ with a linear layer. After that, we replicated the Multihead Attention Layer described in Kool *et al.*[10] using a Graph Attention Network (GAT) [4]. Thus, we apply one GAT layer such that:

$$\mathbf{x_v}' = \mu_{v,v} \mathbf{\Theta} \mathbf{x_v} + \sum_{u \in \mathcal{N}(v)} \mu_{v,u} \mathbf{\Theta} \mathbf{x_u} \qquad (13)$$

with $\mu$ defined by:

$$\mu_{v,u} = \frac{\exp(\text{LeakyReLU}(\mathbf{a}^\top[\boldsymbol{\Theta}\mathbf{x_v}||\boldsymbol{\Theta}\mathbf{x_u}]))}{\sum_{k\in\mathcal{N}(v)\cup\{v\}}\exp(\text{LeakyReLU}(\mathbf{a}^\top[\boldsymbol{\Theta}\mathbf{x_v}||\boldsymbol{\Theta}\mathbf{x_k}]))}, \tag{14}$$

where $\mathbf{a} \in \mathbb{R}^{2\times d_v}$ and $\boldsymbol{\Theta} \in \mathbb{R}^{d_v\times d_e}$ are the trainable parameters. Here, $d_e$ is the original embedding dimension of $\mathbf{x_v}$, $d_v$ is the dimension of $\mathbf{x_v}'$. We apply these equations with $n_h$ different $\boldsymbol{\Theta}$ and $\mathbf{a}$, $n_h$ being the number of heads used in the attention layer. Then, we project back in $\mathbb{R}^{d_e}$ the $\mathbf{x_v}'$ with a linear layer, and sum the $n_h$ resulting vectors. After that, we apply a skip connection [6] and a Batch-Normalization [7] layer BN such that:

$$\mathbf{x_v}' = \text{BN}(\mathbf{x_v} + \mathbf{x_v}'). \tag{15}$$

Finally, we introduce a feedforward network FF which is a 2-layer fully connected network with ReLU activation functions. The input and output dimensions are $d_e$ and the hidden dimension is $d_h$. The final output is then:

$$\mathbf{x_v} = \text{BN}(\mathbf{x_v}' + \text{FF}(\mathbf{x_v}')). \tag{16}$$

We repeated the process described between equation (13) and equation (16) a total of $n_a$ times. Then, to propagate the information of each node to the others in the same connected component, we use an APPNP layer [9]. Given the matrix of nodes embedding $\mathbf{X}^{(0)}$, the adjacency matrix with inserted self-loops $\hat{\mathbf{A}}$, $\hat{\mathbf{D}}$ its corresponding diagonal degree matrix, and a coefficient $\alpha \in [0,1]$, it recursively applies $K$ times:

$$\mathbf{X}^{(k)} = (1-\alpha)\hat{\mathbf{D}}^{-1/2}\hat{\mathbf{A}}\hat{\mathbf{D}}^{-1/2}\mathbf{X}^{(k-1)} + \alpha\mathbf{X}^{(0)}. \tag{17}$$

We used $K = 23$ when we trained on instances from $\mathbb{D}^{(1)}$ and $K = 60$ when we trained on $\mathbb{D}^{(2)}$.

**Graph embedding .** Given the resulting nodes embedding $\mathbf{x_v} \in \mathbb{R}^{d_e}$, in a skip-connection fashion, we concatenate the $\mathbf{x_v}$ back with the original two features $(w'_v, \mathbb{1}_{v\in I})$ ($w'_v$ *being the normalized weights*). Then, the graph level representation vector is, for a graph of size $n$:

$$\mathbf{r} = \sum_{i=1}^{n}\text{softmax}(h_{gate}(\mathbf{x_i})) \odot h_r(\mathbf{x_i}). \tag{18}$$

Here, $h_{gate}$ and $h_r$ are feedforward neural networks with 2 layers and using ReLU activation functions. For both, the input dimension is $d_e + 2$ and the hidden dimension is $d_h$. For $h_r$ the output dimension is $d_e$ whereas for $h_{gate}$, it is 1. We used $n_p$ different versions of the parameters and concatenated the $n_p$ different outputs such that the final graph embedding has a dimension of $n_p \times d_e$.

**Final steps .** We now have the nodes embedding $\mathbf{x_v} \in \mathbb{R}^{d_e}$ and a graph representation $\mathbf{r}$ of dimension $d_e \times n_p$. But the context for each node is not entirely contained in $\mathbf{r}$: the budgets, the size of the graph $n$ and the total sum of weights in the graph are still missing. Thus, we form a context vector $c_o$ as follows:

$$c_o = \mathbf{r}||(n, \Omega, \Phi, \Lambda, \Omega/n, \Phi/n, \Lambda/n, \sum_{v\in V}w_v). \tag{19}$$

When this is done, we perform, for each node, the concatenation $\mathbf{x_v}||c_o$. This is the entry of a feedforward neural network, $\text{FF}_V$ or $\text{FF}_Q$, that computes, for $\hat{V}$, the probability of each node being saved given the context, and the state-action values for $\hat{Q}$. The two feedforward networks are 3-layers deep, with the first hidden dimension being $d_h$ and the second $d_e$. We used LeakyReLU activation functions, Batch Norm and dropout [13] with parameter $p$. Indeed, our experiment shows that using dropout at this stage helps prevent overfitting, and Batch Norm speedups the training. The last activation function for $\text{FF}_Q$ is ReLU whereas for $\text{FF}_V$ we use a sigmoid. Finally, for $\text{FF}_V$, we output:

$$\hat{V}(s) = \sum_{v\in V}P(v\text{ is saved}\mid\text{context})w_v. \tag{20}$$

For $\text{FF}_Q$, we just mask the actions not available, *i.e.* the nodes that are already labeled as attacked.

**Hyperparameters .** All the negative slopes in the LeakyReLU we used were set by default at $0.2$. The value of all the other hyperparameters we introduced here were fixed using Optuna [1] with a TPE sampler and a Median pruner. The objective we defined was the value of the loss of $\hat{V}$ on a test set of exactly solved instances $\in \mathbb{D}^{(1)}$ with budgets $\Omega = 0$, $\Phi = 1$, $\Lambda \in [\![0, 3]\!]$. After running Optuna for 100 trials, we fixed the following values for the hyperparameters: $d_e = 200$, $d_h = 400$, $d_v = 100$, $\alpha = 0.2$, $p = 0.2$, $n_a = 7$, $n_h = 3$, $n_p = 3$. It represents a total of $2, 8$ million parameters to train for both $\hat{V}$ and $\hat{Q}$.

## D.2 Parameters of the training algorithms

**Comparison between the** $3$ **algorithms .** For all $3$ algorithms, we used roughly the same values of parameters in order to make the comparison fair. All three algorithms were compared on instances from $\mathbb{D}^{(1)}$. The batch size was fixed to $m = 256$. Although we share our training times for the sake of transparency and to compare the methods, we want to highlight that our code is hardly optimized and that cutting the times presented here may be easy with a few improvements.

For MultiL-Cur, we used a training set of size $100\,000$ and a validation set of $1000$ instances at each stage of the curriculum. As there are 8 distributions to learn from *(as we use afterstates, there is no need to learn the values of instances having $\Omega = 3$, $\Phi = 3$, $\Lambda = 3$ as budgets)*, this amounts for a total of $808\,000$ episodes. At each stage $j$, we trained our expert $\hat{V}_j$ for 120 epochs, meaning that we used a total of $375\,000$ training steps to finish the curriculum, which necessitated a total of 36 hours. Most of the training time was directed towards generating the training sets, *i.e.* performing the greedy rollouts. Moreover, cutting a few hours in this training time is also possible if we do not monitor the evolution of the training on the test sets *(computing the loss on the test sets regularly takes time)*.

For MultiL-MC, we fixed $\mathcal{C}$, the capacity of the replay memory to be equal to $27 \times 256$ so that each Monte-Carlo sample is exactly seen 27 times. We used a total of $700\,000$ episodes here, resulting in an average of $377\,000$ training steps, which took 56 hours. Indeed, the length of the episodes here is longer on average than the ones used in the curriculum as we directly begin from instances sampled from $\mathbb{D}^{(1)}$ and not the ones where moves were already performed randomly. So the rollout process lasts longer, which is what takes time in our algorithm.

Finally, for MultiL-DQN, we used a capacity $\mathcal{C} = 10\,240$. In order to perform the same number of training steps for the same number of episodes than the other two algorithms, we generated our data in batches of size of 16: at each time step, there are 16 new instances pushed in memory. We used a total of $16 \times 60\,000 = 960\,000$ episodes. The number of training steps performed was $370\,000$ on average. The time necessary for that was 29 hours. Although this is lower than the other two methods *(due to a much quicker rollout)*, the optimality gap and approximation ratio were so high *($\eta = 32.55\%$, $\zeta = 1.54$)* with this amount of data that we actually decided to re-launch an experiment using more episodes. The graphs in Table 1 show the behaviour during training of the 3 algorithms with the setting described until now, however the results of optimality gap and approximation ratio for the MultiL-DQN algorithm are those from a different training setting where we used much more episodes. We made a second experiment were we generated batches of size 128 instead of 16, amounting the number of episodes used to $7\,680\,000$ for the same number of training steps. This second experiment took 72 hours, proving that MultiL-DQN actually necessitates way more data than the two other algorithms, for worse results.

For both MultiL-MC and MultiL-DQN, we used a probability $\epsilon$ with an exponential decay: $\epsilon_{start} = 0.9$, $\epsilon_{end} = 0.05$ and a temperature $T_{decay} = 1000$.

**Curriculum on larger graphs .** For the results in Table 2, we trained our experts on $\mathbb{D}^{(2)}$. As these instances are of larger size and theoretically harder to solve, we decided to train for longer our experts $\hat{V}_j$. We used a training set of size $120\,000$, a validation set of size 2000 and a number of epoch per stage of 200 for the MCN problem. For $\text{MCN}_{dir}$ and $\text{MCN}_w$, we used a training set of size $60\,000$, a validation set of size 1000 and 400 epochs at each stage. The MCN training took roughly a week to run, for $\text{MCN}_w$ and $\text{MCN}_{dir}$ it took 5 days: the rollout on larger graphs takes a long time.

## D.3 Details on the test sets

The test sets we used for the results in Table 1 consisted in 1000 exactly solved instances for each of the 8 different training distributions used during the curriculum. For the results in Table 2, we gathered the instances from [2] for the MCN. As they put a threshold of 2 hours for their solver

MCN$^{MIX}$, the number of instances solved for each of the sizes is different. Moreover, an extensive study of the graphs of size $40$ has been done in their paper. For MCN$_{dir}$ and MCN$_w$, we used the solvers described in Appendix C. The size of the training sets considered in Table 2 are then:

Table 3: Sizes of the test sets used.

| | size of $\mathcal{D}_{test}$ | | |
|---|---|---|---|
| $|V|$ | MCN | MCN$_{dir}$ | MCN$_w$ |
| 20 | 120 | 36 | 36 |
| 40 | 876 | 35 | 34 |
| 60 | 110 | 23 | 29 |
| 80 | 101 | 12 | 30 |
| 100 | 85 | 11 | 27 |

# E    Extended Results

## E.1    Training the Q network with more data

As discussed earlier, we trained an agent on $\mathbb{D}^{(1)}$ with MultiL-DQN using two configurations. First, we used $960\,000$ episode for $370\,000$ optimization steps. Faced with the poor results, we re-trained our agent using more data: $7\,680\,000$ episodes for the same number of steps. We compare the results of the two methods in Table 4. We clearly see that training with more data radically impacts the

Table 4: Comparison between two configurations of training for $\hat{Q}$. In Config. 1, we trained with $960\,000$ episodes while in Config. 2 we used $7\,680\,000$. We display the evolution of the losses during training on $8$ test sets of size $1000$. We measure the resulting optimality gap $\eta$ and approximation ratio $\zeta$ on $3$ different test sets, one for each of the $3$ levels of the problem.

| Level | $\eta(\%)$ | $\zeta$ | $\eta(\%)$ | $\zeta$ |
|---|---|---|---|---|
| Vaccination | 29.8 | 1.54 | **6.7** | **1.08** |
| Attack | 35.8 | 1.45 | **21.2** | **1.18** |
| Protection | 28.8 | 1.63 | **4.01** | **1.07** |

results. More than that, there is a necessity of training with many episodes to obtain reasonable results. We also notice a worse behaviour at the attack stage compared to the other two where it is the defender's turn to play. Thus, we may benefit from adapting the MultiL-DQN algorithms to use two Q networks, one for each player.

## E.2    Training the Value network with less data

In order to assess the capacity of our curriculum to use less data and less training steps, we trained our value network on $\mathbb{D}^{(1)}$ using a second configuration. We re-trained our experts using $50\,000$ instances in the training sets, with $60$ epochs at each stage, instead of $100\,000$ and $120$ originally.

Table 5: Comparison between two configurations of curriculum for $\hat{V}$. In Config. 1, we trained with a total of $800\ 000$ episodes and $375\ 000$ optimization steps while in Config. 2 we used $400\ 000$ episodes and $93\ 750$ steps. We display the evolution of the losses during training on 8 test sets of size $1000$ arriving at different stages of the curriculum. We measure the resulting optimality gap $\eta$ and approximation ratio $\zeta$ on 3 different test sets, one for each of the 3 levels of the problem.

| Level | $\eta(\%)$ | $\zeta$ | $\eta(\%)$ | $\zeta$ |
|---|---|---|---|---|
| Vaccination | **0.955** | **1.011** | 1.126 | 1.013 |
| Attack | **0.409** | **1.004** | 0.913 | 1.009 |
| Protection | **0.005** | **1.000** | 0.005 | 1.000 |

The results in Table 5 clearly show that training with half the data and a quarter of the steps in the curriculum hardly affects the end results, demonstrating the efficiency of the method. Training with Config. 2 took $15$ hours compared to the $36$ necessary with Config. 1.

### E.3   Comparing the difficulty to learn to solve the 3 problems

In this part, we propose to compare the difficulty our curriculum has on learning to solve the 3 different problems MCN, $\text{MCN}_{dir}$ and $\text{MCN}_w$ on instances from $\mathbb{D}^{(1)}$. For that, we ran our curriculum in exactly the same way 3 times, except for the distribution of graphs from which we sampled our instances: undirected with unit weights for the MCN, directed with unit weights for $\text{MCN}_{dir}$ and undirected with integer weights for $\text{MCN}_w$. In Figure 3, we compare the values of the 3 validation losses during the training, along with the values of the approximation ratio $\zeta$ and optimality gap $\eta$ on 3 test sets of 9000 exactly solved instances from $\mathbb{D}^{(1)}$ in Table 6.

| Problem | $\eta(\%)$ | $\zeta$ |
|---|---|---|
| $\text{MCN}_w$ | 7.08 | 1.069 |
| $\text{MCN}_{dir}$ | 2.84 | 1.032 |
| MCN | 0.51 | 1.006 |

Table 6: Values of the approximation ratio and optimality gap on a test sets of exactly solved instances from $\mathbb{D}^{(1)}$ for each of the 3 problems.

Figure 3: Evolution of the loss on the successive validation sets during the curriculum for the 3 problem considered.

Both the table and the figure seem to tell the same story: the easiest problem to learn to solve with our curriculum is the MCN, followed by $\text{MCN}_{dir}$, the hardest one being $\text{MCN}_w$.

### E.4 Assessing the ability to generalize to larger graphs

Previous work on learning to solve single level combinatorial problems with graph neural networks such as [5, 10] reported that their trained agent managed to satisfyingly solve instances with larger graphs at test time than the ones used in their training distributions. In order to assess if this holds for agents trained with our curriculum on the multilevel combinatorial problem, we trained, for each of the 3 problems, our agents on both $\mathbb{D}^{(1)}$ and $\mathbb{D}^{(2)}$, then measured how well they behaved on increasingly larger graphs at test time. We report our results in Table 7.

Table 7: Evolution of the optimality gap $\eta$ and the approximation ratio $\zeta$ with the size of the graphs at test time for each of the 3 problems considered.

We clearly see in Table 7 that the experts trained on $\mathbb{D}^{(2)}$, *i.e.* on larger graphs, perform better than the ones trained on $\mathbb{D}^{(1)}$. From the curves, it seems that our experts can generalize to graphs up to 2 times larger than the ones they were trained on. The fact that for the curves about $\mathbb{D}^{(1)}$ there is first an increase of the values of the metrics and a sudden decrease around $|V| = 80$ may be explained by the fact that $\eta$ and $\zeta$ do not directly measure the goodness of our heuristics. Indeed, if we were to measure how good the decisions taken at a certain level are, we should solve to optimality the subsequent lower levels, which is not what we do here: we use our heuristics everywhere. Thus, when our heuristics perform too badly at *each* level, *i.e.* defending poorly *but also* attacking poorly, there is a chance that the *value* measured in the end of the game is actually not too far from the one

that would have followed optimal decisions. To produce the graphs in Table 7, we generated 3 test sets, one for each problem, using the solver described in Appendix C with IBM ILOG CPLEX 12.9.0. The number of instances in those datasets for each value of $|V|$ are listed in Table 8:

Table 8: Sizes of the test sets used for the results in Table 7.

| $|V| =$ | 20 | 25 | 30 | 35 | 40 | 45 | 50 | 55 | 60 | 65 | 70 | 75 | 80 | 85 | 90 | 95 | 100 |
|---|---|---|---|---|---|---|---|---|---|---|---|---|---|---|---|---|---|
| $\text{MCN}_w$ | 36 | 36 | 36 | 35 | 34 | 33 | 29 | 29 | 29 | 28 | 31 | 29 | 30 | 28 | 29 | 29 | 27 |
| $\text{MCN}_{dir}$ | 36 | 36 | 36 | 34 | 35 | 32 | 29 | 27 | 23 | 17 | 13 | - | 12 | - | 10 | - | 11 |
| MCN | 36 | 36 | 36 | 32 | 30 | 27 | 29 | 26 | 24 | 22 | 17 | 15 | 14 | - | 9 | - | 10 |

## E.5  Identifying multiple optimal solutions

In many situations, there exists multiple solutions to an instance of a combinatorial problem. For some methods, this can represent a challenge as it clouds the decision-making process [11]. However, being able to produce multiple optimal solutions to a combinatorial problem is of interest. Here, the formulation we used naturally allows to identify many of the optimal solutions, assuming our value networks correctly approximate the values of each afterstate. Indeed, if our agents correctly label each node with its value *(i.e. the value of the afterstate if the action is performed on the node, plus reward)*, then identifying all the possible ways of acting optimally is directly readable from them, as shown in the example presented in Figure 4.

(a) Exact values

(b) Approximate values

Figure 4: Exact values and approximate values on an instance of MCN constituted of a graph $G$ and budgets $\Omega = 1$, $\Phi = 1$, $\Lambda = 2$. The exact value of each node is obtained by removing *(vaccinating)* the said node from $G$ and solving exactly the subsequent afterstate with $\Omega$ set to 0. The approximate values are obtained by feeding the afterstates to the expert trained on budgets $\Phi = 1$, $\Lambda \in [\![0, 3]\!]$ during the curriculum for instances from $\mathbb{D}^{(1)}$.

In Figure 4, there are 2 optimal vaccinations: the two blue nodes with value 12. Although the approximate values are not perfectly aligned with the exact ones, the two optimal decisions are clearly identifiable from them, demonstrating the ability of our method to detect multiple optimal solutions.