[Reviews · NeurIPS 2020]

Review 1

Summary and Contributions: The authors phrase the problem of Multilevel Budgeted Combinatorial prob-lems as Alternating Markov Games. A simple Multilevel Q-Learning algorithmis extended to a Multilevel Curriculum learning algorithm that leverages onthe combinatorial setting with a-priori available sequence-lengths, by iterativelygenerating harder training sets. They are used to train a graph neural network-based agent, using copies of early training stages as experts for later training.

Strengths: The authors present a novel and non-trivial approach for multi-level budgeted combinatorial problems. The writeup is clear and the experimental results are promising. The topic is not mainstream for the NeurIPS community but still relevant enough as the connections with discrete optimization are on the rise.

Weaknesses: The addressed problem does not have established benchmarks so it is hard to evaluate the real strength of the method. The main component -- the curriculum learning -- is strongly inspired by Bengio et al, which puts the novelty into question to some extent. The performance on directed and weighted graphs (Table 2) is not compared to DAAD. Is there a reason for it? The overall architecture is somewhat complex (given that it is applied on a single task). Are all components necessary? Maybe some ablations would be helpful.

Correctness: The claims seem correct to me. Experimental methodology makes sense to me.

Clarity: Yes, the write-up is very clear.

Relation to Prior Work: Yes, it is sufficiently discussed. One detail on line 109. The MuZero, AlphaZero extensions of AlphaGo are not committed to a single game.

Reproducibility: Yes

Additional Feedback: line 104: What if player 2 always does nothing? Maybe one could compare to a single-player multilevel task as well? Not sure if there is an application or baseline for that, also not really the point of the paper line 112: typo line 139: For clarification: emphasize how budgets at different timesteps look like e.g. $(0, 0, 0, k, b_{l+1}^t, ..., b_L^t)$ \item equation 8: Again for clarification instead of $... \circ N_{1,\pi^*}^{b_1^\text{max}}$ use $... \circ N_{1,\pi^*}^{b_1^\text{max}-1} \circ N_{1,\pi^*}^{b_1^\text{max}}$ Algorithm 1: although it is quite obvious maybe add that on evaluation the $y_k's$ are sampled from $D_\text{val}^j$ line 208 appendix: typo === POST REBUTTAL === This is just to indicate that I read the rebuttal as well as the other reviews. My evaluation stands.


Review 2

Summary and Contributions: UPDATE AFTER AUTHOR RESPONSE: I thank the authors for the response. Based on your clarifications I have increased my score. ================================== The paper tackles the problem of multi-level combinatorial optimization, formulated as a two-agent, zero-sum, turn-based game. The authors formulate this as a reinforcement learning problem and propose an algorithm that uses curriculum learning to solve it. The proposed solution seems to bring some benefits in terms of the data and number of steps needed to solve the problem for relatively small graphs (<100 nodes), but does not scale beyond this. Regardless, it provides a good first step that could inspire more future curriculum work in the area.

Strengths: 1. Well written introduction, with compelling arguments for the importance of the problem, with applications to security, flood control, matrix decomposition, etc. 2. The idea of using curriculum learning in this setting makes sense, and it is a good idea to bring it up to the combinatorial optimization and game theory community. 3. The results look promising, at least for small graphs.

Weaknesses: 1. Related work on curriculum learning is missing. 2. I found section 4.4. that discusses the proposed curriculum algorithm pretty difficult to read (see more details below). 3. The current version of the curriculum algorithm seems to have some scalability limitations.

Correctness: As far as I can tell, the experiments seem to be conducted correctly, and the authors provide plenty of details in the appendix that should help with reproducing them.

Clarity: Up to section 4.4. I found it very easy to follow. The introduction is particularly well written and motivated. Moreover, the use of English is flawless. In section 4.4. things become more complicated to follow. I’m not sure I understood why in equation 7 B_t = (0, .., 0, k, b_{l+1}, .., b_L). In general, eq. 7 could use more explanation, as well as algorithm 1.

Relation to Prior Work: The discussed work is clearly explained, but I cannot vouch for completeness on the combinatorial optimization literature side, as I am not very familiar. However, the curriculum learning literature is not discussed at all in the intro or related work section, and I think it should be. There is a brief comparison with the work of Bengio et. al in section 4.4., but this paper is more than 10 years old and dozens of other methods have been proposed since then, especially a lot of approaches for reinforcement learning.

Reproducibility: Yes

Additional Feedback: Questions: 1. How do you set T_val in Algo 1? 2. Given the constraints added in the problem description (e.g., no incentive to pass, perfect information, etc), I am wondering what are some real-world problems that fit this constrained problem formulation? Suggestions: 1. Add a comparison with curriculum learning literature. Is the curriculum side of the proposed method novel in any way? Could it provide insights for other problems in combinatorial optimization? 2. Introduce section 5 earlier, and use it as a running example in Section 4, it would help a lot in understanding it more intuitively. Corrections: I noticed something funny with the bib file. There is a citation in the main text “Dai et al. [30]” but if you look at the list of authors of paper [30], it seems that Dai is the second author. I’m wondering if that’s an issue with your bib file or it was intentional.


Review 3

Summary and Contributions: This paper proposes a deep reinforcement learning approach to solve multi-level budgeted combinatorial problems. To this end, the authors propose to formulate a multi-level budgeted combinatorial problem as an alternating Markov game. While the proposed alternating Markov game can be solved with regular deep reinforcement learning, the authors propose an additional curriculum learning scheme that boosts the training of agents for solving the multi-level budgeted combinatorial problem. Experiments on the multilevel critical node problem validate the performance of the proposed schemes.

Strengths: - This paper newly proposes the multi-level budgeted combinatorial problem with deep reinforcement learning, which is a novel combination of task and methodology to my knowledge. - Overall, the proposed methodology is very reasonable and sound. - The proposed curriculum learning is delivers an interesting message (later stages of the MDP are easier to train). - Empirical evaluation is complete with details.

Weaknesses: - Experiments were done only on small-scale synthetic datasets. Hence it is unclear whether the proposed method is useful in real life. - While the paper targets general multi-level budgeted combinatorial problems, experiments were conducted only on the three-level critical node problem. (Is it also useful for higher-levels?) - Table 2. hints that the proposed approach does not perform well for large-scale instances, e.g., when |V|=100. I think this should be investigated further, e.g., trying the experiment for |V|=200. - The proposed curriculum learning strategy delivers a similar message made by Florensa et al. 2017, where a robot is trained in reverse, gradually learning to reach the goal from a set of start states increasingly far from the goal. It would be useful if the authors could discuss this work. Florensa et al. 2017, Reverse Curriculum Generation for Reinforcement Learning.

Correctness: The proposed methodology seems reasonable and the propositions are correct to my knowledge.

Clarity: This paper was very clear and easy to read.

Relation to Prior Work: The positioning of the paper can be improved by making reference to existing works that use curriculum learning for reinforcement learning, e.g., [Florensa et al. 2017].

Reproducibility: Yes

Additional Feedback: - Overall, this paper could be improved by showing its significance in practical and realistic scenarios. - I think Section E.4 (in the appendix) is worth mentioning in the main text since it addresses an important case where the solver is evaluated on distribution unseen during training. - To my knowledge, rollout using only the V function is more costly than Q function. It requires the value network to evaluate every possible future state (which has a higher dimension than the action). I am concerned that this approach might not be scalable to large graphs. It would be useful if the authors could elaborate more on this aspect. - It seems that the original MCN paper [Baggio et al. 2018] also considered a tree-structured graph in the experiments. It would be useful if the authors also experiment in this case. ============ I appreciate the author's thoughtful and detailed rebuttal. I have changed my score since the rebuttal persuaded me on (a) the significance of tackling a new combinatorial optimization problem with deep RL and (b) novelty of the proposed curriculum learning compared to prior works. I am still concerned that this paper does not provide enough empirical evidence for the proposed algorithm to be useful in practice. However, this seems fine given that the proposed problem itself is in an early stage of development.


Review 4

Summary and Contributions: This paper proposed a curriculum-based framework to solve a multilevel combinatorial optimization problem with a budget. The authors formulated the problem as multi-agent reinforcement learning(RL) and proposed RL-based algorithms. They conducted experiments on the Multilevel Critical Node problem (MCN) and compared their results with exact and heuristic-based solvers.

Strengths: This paper addresses a problem which is an interesting use case of RL. The paper is overall well-written and the methodology is well-explained. The proposed method sounds novel and introduces a multi-agent RL framework into combinatorial optimization problems. Thus this paper brings an interesting perspective relevant to the NeurIPS community.

Weaknesses: The significance of the results compared to baselines needs to be explained and justified. Some details are missing which are necessary to understand the significance of the proposed framework. Please refer to the details comment section.

Correctness: Claims and methodology seem correct. However, I have a concern about results comparison, time(s) metric. I am expecting an explanation from the authors. Please refer to the details comment section.

Clarity: Overall, the paper is well written and easy to follow. There are a few points which need further clarification. They are in the comment section.

Relation to Prior Work: Yes.

Reproducibility: Yes

Additional Feedback: While I do think this is a good contribution, I have the following concerns which, if resolved, will help me to champion this paper. Table 1: The loss values are on different scales thus they are not comparable. Any reward-based metric (average reward) might mitigate this issue. Table 2: DA-AD results are from the reference paper [1]. A comparison of computing resource configuration is needed for the time comparison. The reference paper [1] uses a machine configuration “Intel Xeon E5-2637 processor clocked at 3.50GHz and 8 GB RAM, using a single core”. While this paper uses a different machine “one gpu of a cluster of NVidia V100SXM2 with 16G of memory”. An explanation is necessary to understand the performance improvement in the proposed method. How much improvement because of computing resources and how much due to the proposed technique. Results in Table 2: The difference between DA-AD and cur in optimality gaps, and approximation ratios seem small in number. How significant are these differences? Any statistical measure or empirical justification would strengthen the proposed improvement. Does the training data consist of exactly solved instances? What dataset is used for the results in Table 2? Is it D^1 or D^2? What is the train test split in the results in Table 2? It seems the testing inference time is relatively quick (few seconds) for up to 100 sized graphs. It would be interesting to see how they transfer to the larger graphs, for example, graphs with #node =1000, 5000, etc. Larger graphs are common in many real-world networks (e.g., social media). The bottleneck seems to be the training time (e.g., MultiL-Cur) as it is not scaled well with graph size. Algorithm 1: Line 3: what is the rationale to use 2? Will the loop run at most 2 times? What is the value range of B_max? What is “number of new updates” and T_val? How are they calculated? Comments can be added in the algorithm for such clarification. Section 5: Figure 1 is not clear to me, how the graph is constructed in different steps, and also how the values of \Omega, \phi are calculated? A description of them would be easier to follow in Figure 1. How does the proposed framework help to generate heuristics (line 302). Training RL-algorithms take time and training data. An elaborate discussion of how RL can be useful for practical use would be helpful for readers especially when limited or no training data available.

[Author Response · NeurIPS 2020]

We thank the reviewers for their valuable feedback. We are glad that they found our approach novel and promising, and agree that further details would facilitate the understanding of Algo. 1. Next, we answer to the presented comments: positioning in the literature for curriculum learning, scalability of our approach, experiments details, and applicability. Finding ways to automatically design a sequence of learning tasks that are increasingly *harder* to solve for the agent is a challenge in curriculum learning. For *supervised* learning, [7][1] showed that gradually increasing the *entropy* of the training distribution helped. However in RL, breaking down a task in sub-problems that can be ordered by difficulty is non trivial [2]. In robotics, [1, 3] proposed to start from the *goal* (*e.g.*, open a door) and give a starting state that is gradually further from that goal. These methods assume at least one known goal state that is used as a seed for expansion. For video games, [4] adapted the concept with a starting state increasingly further from the end of a *demonstration*. However, here, the goal is not to "reach a particular end state": there is no goal state at all. Rather, what we want is to "take an optimal decision at each time step", and, particularly, take optimal decisions at the beginning of the game, *i.e.*, at the highest level of the multilevel optimization problem, given that all subsequent actions *will* be optimal. Thus, contrary to [1, 3, 4], we do not "reverse time" to artificially build a sequence of tasks starting further from a goal state and subsequently harder to solve in the hope of learning how to reach this goal from all possible starting states, but rather *stack* new optimization problems on top of previous ones, which gradually increases the *computational complexity* of the task, in order to learn to act optimally in optimization problems with an increasing number of levels. For example, in the case of $MCN_w$, solving the last stage (protection) is NP-hard whereas solving the bilevel min-max (attack knowing the subsequent protection will be optimal) is $\Sigma_2^p$-hard, and the trilevel problem (vaccination knowing the attacker will react optimally knowing we will be able to find an optimal protection after) is $\Sigma_3^p$-hard [45]. Thus, contrary to most problems in RL, here we are faced with a task *naturally* constituted of a hierarchy of sub-problems ordered by their position in the Polynomial Hierarchy, which motivates a curriculum. The one we devised is based on an *afterstate value function*, raising scalability issues as mentioned in our discussion: this is exactly what we leave as a future direction. The paper's methodology will benefit its pursuer. So we must stress the impact of this work as the first looking to such problems and setting the ground for less expensive curricula. In Operations Research, most of the multilevel combinatorial problems studied have less than 4 levels in practice: finding exact methods to solve bilevel and trilevel problems is still an active area of research; note that an MBC with $L$ levels is potentially $\Sigma_L^p$-complete. Thus, even-though scaling our method to more levels is straightforward, we did not tested it as finding reference problems for such situations is rare. This also explains the scarcity of benchmarks for trilevel problems and our choice of focusing on the MCN: there is a methodology to solve the problem along with a publicly available dataset of exactly solved instances. We used this dataset to *evaluate* MultiL-Cur in Table 2: these instances were never seen before by our agent. To *train* our agent, we generated our own dataset of instances, and used as targets the *approximate* values given by the `Greedy Rollout` procedure. For the validation, we arbitrarily set $T_{val}$ to a relatively low value of 20. In Table 2, only 71% of the 120 instances of size 100 generated for [1] were solved by $MCN^{MIX}$ under their threshold of 2h; we only considered those as they are the only ones with a solution. But, if we had added a 2h lower bound for each non solved instance in our time average for $MCN^{MIX}$, the entry for graphs of size 100 would report 2690s instead of the 848s. Thus, generating a test set of exactly solved instances of larger size would take time with the existing methods, explaining why we did not try to benchmark the abilities of our heuristic on significantly larger graphs. Finally, in Table 2, we did not compare our curriculum to DA-AD in $MCN_{dir}$ and $MCN_w$ as [1] did not have results on this; we only adapted their exact method (Appendix C). Plus, from the complexity point of view, these cases should be much more expensive for DA-AD [45]. So our goal was to show a meaningful comparison for the simplest case. To gain more hindsight on the metrics $\eta$ and $\zeta$, one can look at Fig.2 of [30] and Fig.5 of [33]. Regarding the usefulness of such problems for practical scenarios, the MCN could fit on several applications, *e.g.* to limit the fake news spread in social networks or in cyber security for the protection of a botnet against malware injections. In the latter, the attacker infects nodes by introducing a malware in some bots, the defender vaccinates and protects nodes by disconnecting them, stopping the spread of the malware.

## Footnotes

[1]citations in blue refer to the bibliography of the paper.

# References

[1] Carlos Florensa, David Held, Markus Wulfmeier, Michael Zhang, and Pieter Abbeel. Reverse curriculum generation for reinforcement learning. *arXiv preprint arXiv:1707.05300*, 2017.

[2] Alex Graves, Marc G. Bellemare, Jacob Menick, Remi Munos, and Koray Kavukcuoglu. Automated curriculum learning for neural networks. *arXiv preprint arXiv:1704.03003*, 2017.

[3] Boris Ivanovic, James Harrison, Apoorva Sharma, Mo Chen, and Marco Pavone. Barc: Backward reachability curriculum for robotic reinforcement learning. *arXiv preprint arXiv:1806.06161*, 2018.

[4] Tim Salimans and Richard Chen. Learning montezuma's revenge from a single demonstration. *arXiv preprint arXiv:1812.03381*, 2018.



[Meta-Review · NeurIPS 2020]

The paper proposes a deep reinforcement learning approach for multi-level combinatorial optimization. Reviewers agree on accepting the paper and recognize its novelty and writing standard. The paper merits publication due to its novelty (combining combinatorial optimization with RL), being well-written and promising experimental results. The rebuttal addressed successfully concerns raised by reviewers. The experimental part of the paper did not convince the reviewers, though, due to small problem sizes. The recommendation for the paper is accept.